# Effects of Enzymatic- and Ultrasound-Assisted Extraction on Physicochemical and Antioxidant Properties of Collagen Hydrolysate Fractions from Alaska Pollack (*Theragra chalcogramma*) Skin

**DOI:** 10.3390/antiox11112112

**Published:** 2022-10-26

**Authors:** Ju Eun Lee, Sang-Kyu Noh, Mi Jeong Kim

**Affiliations:** 1Interdisciplinary Program in Senior Human Ecology, Changwon National University, Changwon 51140, Korea; 2Department of Food and Nutrition, Changwon National University, Changwon 51140, Korea

**Keywords:** Alaska pollock skin, collagen hydrolysate, ultrasound assisted extraction, enzyme assisted extraction, antioxidant properties, microstructure

## Abstract

Collagen hydrolysate were extracted from Alaska pollock skin using enzymatic (EAE), ultrasound (UAE), or combination of enzymatic and ultrasound (EAE+UAE) treatment. Control (C) was not treated with enzymatic or ultrasound. The extracts from C, EAE, UAE, and EAE+UAE were fractionated with ≤3, 3–10, 10–30, and ≥30 kDa. Each fraction was evaluated for biological activity and structural properties. All fractions contained high levels of glycine and proline. The ≤3 kDa fraction of control and ultrasound-assisted extracts exhibited the highest antioxidant activity as measured using Trolox equivalent antioxidant capacity, ferric ion reducing antioxidant power, oxygen radical absorbance capacity, and an assay on the inhibition of nitric oxide production by LPS-induced macrophages. The structurally digested collagen was evaluated using FTIR spectra and SDS-PAGE after Alcalase^®^ and ultrasound treatments. The microstructure of collagen hydrolysate was assessed using SEM microscopy; the surface morphology was altered according to fraction size and extraction conditions. Overall, it was determined that enzyme treatment in combination with ultrasound is the most effective procedure for obtaining digested collagen hydrolysate, which could be used to further improve biotechnological processing for the addition of value to marine production chains in the future.

## 1. Introduction

Global fish production amounted to 179 million tons in 2018 and it is expected to increase to 204 million tons by 2030 [1]. Meat by-products accounted for approximately 25% of the total fish catch worldwide, of which only 13% was processed for fishmeal obtention. This implies that the rest was discarded as waste, which often has negative environmental impacts [2]. Numerous studies have suggested that fish by-products might be used as source material for obtaining collagen or collagen hydrolysate to add value to the marine production chain [3,4]. The demand for collagen products in the global food industry is increasing due to the growing interest in the potential health benefits of collagen and collagen peptides, especially in recent years [5].

These proteins have been reported to have major health benefits, including antioxidant effects, matrix metalloproteinase-2 (MMP-2) suppression, angiotensin I-converting enzyme (ACE)-inhibition, and anti-aging activity [6,7,8,9]. These health-promoting properties have been associated with bioactive peptides consisting of amino acid residues [10]. It has been reported that collagen hydrolysates, which have lower molecular weights following hydrolysis, have higher bioactivity and absorption rates than non-hydrolyzed collagen [11]. Several studies reported that low-molecular weight collagen has more significant effects on antioxidant activity [12], antibacterial properties [13], and ACE-inhibitory activity [14] than high-molecular weight collagen. Accordingly [11,12,13,14], low-molecular weight peptides are preferentially produced and consumed in the collagen industry [5]. Collagen hydrolysis on an industrial scale is therefore an economically important technology to produce valuable low-molecular weight collagen hydrolysate. For collagen hydrolysates, extracted collagen can generally be hydrolyzed into low molecular weight peptides using chemical such as acids and alkalis, or enzymatic hydrolysis [5]. Protein hydrolysis by enzymatic treatment might be safer, cheaper, and more environmentally friendly compared to chemical treatments such as acids and alkalis. Enzymatic treatment produces hydrolysate products with reportedly enhanced bioactive and physicochemical properties that improve their anti-cancer and antioxidant effects, and end-product emulsion stability. These properties arise from the differential cleavage of peptides in the polypeptide chain around the active site [15].

Recently, ultrasound assisted extraction has been widely applied to improve the extraction efficiency of compounds from various sources [16,17]. Also, the combined extraction method with acid and ultrasound has been reported in studies of [6,16,18]. Other researchers reported that protein hydrolysis by Flavourzyme^®^ or pepsin enhanced by the use of previous or simultaneous ultrasound [19,20]. However, there is no scientific information to study the characteristics of the collagen hydrolysate extracted on the combination of enzyme and ultrasound by molecular size fractionation. Ultrasonic treatment and enzymatic extraction are both considered to be a “green” extractive technique that effectively improve extraction yield of collagen hydrolysate from fish skin structures. Also, collagen hydrolysates extracted from fish skin might be existed with various molecular weight peptides and various activities of collagen hydrolysate peptides differ depending on the molecular weight [10]. Therefore, this study aimed to measure the antioxidant and structural characterizations of collagen extracted using enzyme and ultrasound processing by comparing four fractions (1; ≤3 kDa, 2; 3–10 kDa, 3; 10–30 kDa, 4; ≥30 kDa) molecular weight. 

## 2. Materials and Methods

### 2.1. Preparation of Collagen Hydrolysate

The collagen hydrolysate from Alaska Pollock skin (APS) was extracted by methods used in the study of Yang and Hong [21] with slight modifications. For collagen extraction, dried APS was made to a size of 0.25 cm^2^ and soaked in distilled water for 1 h. The water drained off and 0.1 N NaOH was added to the APS. The APS was gently stirred with 100 rpm at cold room of 4 °C for 24 h. The alkali-pretreated APS was neutralized by washing with distilled water until the pH value reached 7. To collect APS collagen extract, the neutralized samples were extracted in a shaking incubator (Hankuk Scientific Instrument Company, HK-SI25C, Hwaseong-si, Korea) at 55 °C with a shaking frequency of 100 rpm for 24 h.

The APS collagen extract was hydrolyzed using Alcalase^®^ 2.4 AU/g enzyme solution purchased from Novozyme (Novozyme Nordisk, Bagsvaerd, Denmark) and an ultrasound processor (Omni Sonic Ruptor 400, Omni, Kennesaw, GA, USA). For enzyme assisted extraction, 1% Alcalase^®^ solution was added to the neutralized APS samples, and then hydrolyzed in a shaking incubator (Hankuk Scientific Instrument Company, HK-SI25C, Hwaseong-si, Korea) at 55 °C at 100 rpm for 24 h. After hydrolysis, Alcalase^®^ enzyme activity was stopped by heating at 85 °C for 20 min. Ultrasound assisted extraction was conducted on APS collagen extract or collagen hydrolysate with Alcalase^®^ using an ultrasound processor (Omni Sonic Ruptor 400, Omni, Kennesaw, GA, USA) with a probe at 20 kHz (80% amplitude) for 30 min. 

Each APS collagen extract or hydrolysate sample was first filtered with Whatman filter paper No. 4, and centrifuged at 8000× *g* for 10 min. Each supernatant was fractionated using 3, 10, and 30 kDa centrifugal filter units (Amicon^®^ Ultra Centrifugal Filters, Millipore, Burlington, MA, USA) to fractionate molecules with different molecular weights. Each fraction of APS collagen extract and hydrolysate was lyophilized with a freeze dryer (Bondiro, Ilshinbiobase, Gyeonggi-do, Korea). The condition of a freeze dryer was set with a condenser temperature of −70 °C and chamber pressure of 0.02 mbar. 

### 2.2. Yield of Collagen Hydrolysate

The yield of collagen hydrolysate fractionated with ≤3, 3–10, 10–30, ≥30 kDa filters was calculated as described by Ali, et al. [22] using the following equation:(1)$$Yield\;\left(\%\right)=\frac{Weight\;of\;lyophilized\;collagen\;hydrolysate\;}{Initial\;weight\;of\;dried\;APS}\times 100\;$$

### 2.3. Antioxidant Properties

#### 2.3.1. Trolox Equivalent Antioxidant Capacity (TEAC)

The TEAC assay measured scavenging of ABTS radicals according to the method of Li, et al. [23]. The radical solutions were produced by adding 7.4 mM ABTS and 2.6 mM potassium persulfate in a 1:1 ratio. The solution was allowed to react in a darkroom at 25 °C for 16 h. The ABTS^∙^ working solution was prepared with ethanol until an absorbance of 0.700 ± 0.01 at 734 nm. The 2960 µL of diluted working solution was mixed with 40 μL of collagen hydrolysate. The mixture was allowed to react for 7 min and was then measured at 734 nm using a microplate reader (EPOCH 2, Biotek, Winooski, VT, USA). Trolox was used as a standard. The results were expressed as μmol of Trolox equivalents (TE) per gram of lyophilized collagen hydrolysate. 

#### 2.3.2. Ferric Ion Reducing Antioxidant Power (FRAP)

The FRAP assay was conducted by the method developed by Chen, et al. [24]. The FRAP solution was prepared by mixing 2.5 mL of 10 mmol/L TPTZ (2,4,6-Tripyridyl-*s*-triazine in 40 mmol/L HCl), 25 mL of 300 mM acetate buffer (pH 3.6) and 2.5 mL of 20 mmol/L FeCl_3_ in distilled water and warming the solution to 37 °C before use. Samples were prepared as 0.1 mL aliquots (0.01 g/mL) and were reacted with 2.4 mL of FRAP solution at 37 °C in a darkroom for 10 min. The mixture was measured at 593 nm using a microplate reader (EPOCH 2, Biotek, Winooski, VT, USA). Ferrous sulfate (FeSO_4_·7H_2_O) was used as a standard and was expressed as milligram of FeSO_4_ equivalent per gram of lyophilized collagen hydrolysate. 

#### 2.3.3. Oxygen Radical Absorbance Capacity (ORAC) 

The ORAC was determined using the method described by Kim and Kim [25]. A solution of fluorescein and AAPH was made using 75 mM phosphate buffer (pH 7.4). 25 μL of collagen hydrolysate (0.01 g/mL) and 150 μL of fluorescein were transferred into a 96-well black plate and was pre-heated at 37 °C before reaction. The reaction was started by adding 25 μL of 79.6 μM AAPH, and absorbance was measured every 1 min for 1 h with excitation and emission wavelengths of 485 nm and 538 nm, respectively. The ORAC value of each sample was expressed as μmol Trolox equivalents (TE) per gram of lyophilized collagen hydrolysate.

### 2.4. Nitrite Oxide Production

The nitric oxide (NO) content was determined with methods reported by Song et al. [26]. RAW 264.7 macrophage cells were cultured overnight at 1 × 10^5^ cells per well in a 48 well plate (SPL Life Sciences, Pocheon, Korea). Cells were then incubated with APS collagen hydrolysate (1 mg/mL) and LPS (1 μL/mL) for 24 h. Thereafter, 50 μL of the resulting supernatant and 50 μL of Griess reagent (Sigma-Aldrich, St. Louis, MO, USA) were reacted for 10 min and were then measured at 540 nm with a microplate reader (EPOCH 2, Biotek, Winooski, VT, USA). A standard calibration curve was constructed using sodium nitrite as a standard, and the nitric oxide content of each sample was calculated.

### 2.5. Scanning Electron Microscopy 

The surface morphology of lyophilized collagen hydrolysate was observed with Low voltage scanning electron microscopy (LV-SEM) (Merlin compact, Zeiss, Oberkochen, Germany) at a voltage of 15 kV. Each sample was attached to adhesive carbon tape, and a thin layer of gold coating was applied before measurement. The SEM micrographs of the samples were obtained at a magnification of 5000× to investigate changes in the microstructure caused by different extraction techniques.

### 2.6. Fourier Transforms Infrared Spectroscopy (FTIR)

The FTIR spectra were determined by the methods of Sun, et al. [27]. Each lyophilized collagen hydrolysate (1 mg) was mixed with 100 mg dried potassium bromide, and FTIR spectra were scanned using an instrumental resolution of 4000 to 400 cm^−1^ with scanning time of 16 s using an infrared spectrophotometer (FTIR-4200, JASCO, Tokyo, Japan).

### 2.7. Sodium Dodecyl Sulphate-Polyacrylamide Gel Electrophoresis (SDS-PAGE)

The protein pattern of collagen hydrolysate was determined with SDS-PAGE using the methods of LAEMMLI [28]. The samples were dissolved at a ratio of 3:1 with 4× Laemmli sample buffer (Bio-Rad, Hercules, CA, USA) in the presence of 10% (*v*/*v*) βME, and were then heated at 95 °C for 5 min. The heated collagen sample (10 μL) was loaded on polyacrylamide gel made of 10% resolving gel and 4% stacking gel, and electrophoresis was conducted at a constant voltage of 100 V using a Mini-PROTEAN II electrophoresis system (Bio-Rad, Hercules, CA, USA). Then, the gel was stained using Bio-Safe Coomassie stain (Bio-Rad, Hercules, CA, USA) overnight. 

### 2.8. Statistical Analysis

Yield, TEAC, FRAP, ORAC, control, and FTIR analyses were performed in triplicate, and all data were calculated as the mean ± standard deviation. The differences between samples were determined by analysis of variance (ANOVA) using a XLSTAT software 2022 (Addinsoft, Paris, France). The Duncan multiple comparison test was used to assess differences between sample means with significance defined at a *p*-value of <0.001.

## 3. Results and Discussion

### 3.1. Yield of Collagen Hydrolysate

The extraction yield of APS collagen hydrolysates, which were processed with ultrasound only, with enzyme treatment only, with both treatments, and with no treatment is presented in Table 1. The total yields for the control and ultrasound-assisted extract (UAE) were 31.0% and 40.2%, respectively. In presence of Alcalase^®^, the yield was enhanced drastically. The enzyme-assisted extract (EAE) and enzyme combined with ultrasound treatment extraction extract (EAE+UAE) had average yields of 62.7%, and 65.3%, which were approximately 1.5–2 fold higher than that of the control. Enzymes are commonly used to cleave peptide bonds to facilitate extractions in industrial settings. Proteases, including Alcalase^®^, hydrolyze proteins into biologically active peptides or free amino acids. Alcalase^®^ has been previously shown to produce a large quantity of low molecular weight protein hydrolysate, with a hydrolysis capacity that is about 10 fold higher than that of enzymes such as trypsin and papain [29]. Therefore, these experiments serve as further evidence that the application of Alcalase^®^ increases the yield of EAE and EAE+UAE more effectively through the hydrolysis of skin. Furthermore, it is of note that the ultrasound treatment also increased yield compared to the control. The EAE+UAE sample presented the highest yield, and UAE treatment alone showed a yield that was about 9% higher than that of control sample. This means that the yield was increased by ultrasonic extraction in the absence of an enzyme. This supports previous findings, which suggested that ultrasound-assisted extraction improves hydrolysate yield, resulting from the implosion of cavitation bubbles, generation of microjets, and micro-turbulence which disrupt the skin matrix [30]. Collagen from control samples had the lowest yield (31%). 

As for the yield by the fraction size, the control and UAE groups showed the highest yields of ≥30 kDa fragments at 21.16% and 25.99%, respectively. The yield from the EAE and EAE+UAE groups consisted of predominately 10–30 kDa fragments. These results demonstrate that enzyme treatment increases the yield of fragments below 30 kDa and likely lowers the molecular weight of collagen through hydrolysis.

### 3.2. Antioxidant Properties and NO Production

The antioxidant activities of crude extract and APS collagen hydrolysate fractions obtained using various processing methods were evaluated by TEAC, FRAP, and ORAC assays as shown in Figure 1. TEAC and FRAP assay results (Figure 1a,b) showed similar trends in scavenging activity where the control and UAE fragments with low molecular weights (≤3 kDa) had the highest results of all fractions. It is thought that this result is attributable to the comparable redox potentials between Fe^3+^-TPTZ and ABTS^•+^, which are <0.7 V and 0.68 V, respectively [31]. The Trolox equivalent values for TEAC were highest in C1, at 0.56 mM TE/g, and UAE1 at 0.52 mM TE/g. For the FRAP assay, the values of C1 and UAE1 were 1.03 and 0.91 mM TE/g, respectively. However, the antioxidant values of the enzyme-treated samples (EAE and EAE+UAE) were higher than that of the non-enzyme-treated samples without C and UAE ≤ 3 kDa and 3–10 kDa.

The ORAC assay followed the same trends in results as the TEAC and FRAP assays (Figure 1c). The collagen hydrolysate of the control and UAE ≤ 3 kDa fractions significantly reduced peroxyl radicals, showing values for Trolox equivalents of 9.66, and 9.62 μM TE/g, respectively. The values from the ORAC assay were similar, with a range of 8–9.5 μM TE/g. In a previous study, collagen hydrolysate scavenging activity was demonstrated to have an inverse relationship with fragment size [11]. Zamorano-Apodaca et al. reported that low molecular weight collagen hydrolysates had increased interaction with free radicals. The antioxidant activity of collagen hydrolysate is associated with amino acid content and peptide size. It is also influenced by the number of hydrophobic groups present in the molecule; the reduction of these groups through hydrolysis increases protein interactions and therefore biological activity of the molecule. Reactive glycine, glutamic acid, alanine, and aspartic acid residues in collagen hydrolysates of low molecular weight are more exposed, which increases the opportunity for peptide interactions that improve radical stability. The presence of electron donor, or chelating amino acid enhances the antioxidant effects of hydrolysates [10].

The production of NO, an inflammatory mediator, produced by LPS-induced macrophages in the presence or absence of enzyme or ultrasound-treated collagen hydrolysate is shown in Figure 2. The amount of NO produced increased rapidly from 0.8 to 37.5 μM in response to LPS. However, this increase was reduced when the collagen hydrolysate fraction samples were added to LPS-induced RAW 264.7 cells, NO levels were significantly decreased in the presence of ≤3 kDa fragments from groups C and UAE and 3–10 kDa fragments from C. These results followed the trends observed in the TEAC and FRAP assays where NO reduction was observed at the highest levels in response to the same fractions as increased antioxidant activity. These results strongly suggest that the inhibition of NO and increased antioxidant activity are related. NO plays an important role in immune defense mechanisms. However, many inflammatory disorders are linked to excessive NO and cytokine production. In this study, LPS-induced macrophages were treated with low molecular collagen hydrolysate fraction (≤3 kDa) resulting in suppressed NO production. It has been reported that biologically active peptides of various proteins, such as skeletal muscle and ovalbumin, can inhibit cytokines and exert effects on the inflammatory response [10]. Furthermore, the number of amino acids, their sequence, length, charge, hydrophobicity, and the structure of the peptide all affect its immunomodulatory properties. 

### 3.3. Morphology

The microstructure of collagen was different depending on the fractionation and treatment conditions. SEM micrographs of crude extract and collagen fractions with or without enzyme and/or ultrasound treatment are displayed in Figure 3. Crude extract samples from the control and UAE groups were observed to have small pores in their structure. These pores are characteristic structures of collagen. On the contrary, the structure of collagen peptide fragments from crude extract samples of EAE and EAE+UAE were visually degraded compared to the control. The effects of hydrolysis were observed even in the ≥30 kDa fractions for the EAE and EAE+UAE groups, showing more irregular and rough morphology compared to C and UAE samples. The control and UAE ≤ 3 kDa fractions (C1, and UAE1) showed irregular shapes that consist of very small and thin strands compared to higher molecular weight samples because the surface structure is composed of relatively short sequences of amino acid residues. Enzyme extracted samples (EAE1 and EAE+UAE1) displayed a distinct morphology. The collagen fibers were cracked on the surface resulting in a rough texture that was completely fractured in some areas due to enzymatic hydrolysis. The collagen fractions consisting of larger molecule fractions (10–30 kDa and ≥30 kDa), had cracks on the surface that became less pronounced as the fragment size increased. Overall, these results demonstrate that hydrolysis reduces the size and integrity of collagen particles, resulting in irregularities in morphology. The surface morphology and internal microstructure of these reduced particles might dramatically contribute to the physical and functional characteristics of the collagen hydrolysate [32]. 

### 3.4. Fourier Transforms Infrared Spectroscopy (FTIR)

FTIR spectra results for fraction samples by molecular weight of APS collagen hydrolysate extracted using enzyme and/or ultrasound treatment are shown in Figure 4. The amide A, B, I, II, and III bands were observed in the control, UAE, EAE, and EAE+UAE groups, showing characteristic of typical collagen peptide structures. The absorbance of all amide bands for the control and UAE group crude extracts were higher than those of the EAE and EAE+UAE groups, indicating that the control and UAE groups have more bonds in each band. This was also observed in the ≥30 kDa fractions from these groups. These results indicate that enzymatic hydrolysis produces more extensive bond cleavage, resulting in a change in configuration.

Amide A band readings indicate stretching vibrations of free N-H groups which occur within a range of 3400–3440 cm^−1^. Amide B readings were peaked in approximately 2900 cm^−1^ and represent asymmetrical stretching of both ═C-H and –NH^3+^ groups. Amide I, II, and III bands were observed in the range of 1600–1700 cm^−1^, 1500–1600 cm^−1^, and 1200–1300 cm^−1^, respectively [16]. Amide I band readings are associated with stretching vibrations of C=O groups and were recorded in crude extracts at 1651–1654 cm^−1^. The collagen fractions from all groups were found to have amide I band readings, which were lower than those of the crude extract reading from the same group. The difference in wavenumber depends on the amide C=O group located at each Gly-X-Y residue, and hydration is also considered to be a contributing factor to the difference in spectra [33]. Amide II band readings represent N-H bending. Amide II band peaks were recorded at 1539–1542 cm^−1^ in crude extract samples, whereas the wavenumber for the same reading shifted to 1545–1547 cm^−1^ in fraction samples. Readings at lower wavenumbers indicate that a higher proportion of N-H bonds are involved in hydrogen bonding with adjacent α-chains [22]. Amide III is representative of C-N stretching and N-H deformation, which are involved in the complex intermolecular interactions of collagen [16]. The ratio of the peak amplitudes between Amide III and the 1454 cm^−1^ band is approximately 1.0, with the exception of the control ≤3 kDa, UAE ≤ 3 and 3–10 kDa fractions, indicating the presence of a triple helix structure. This might be attributable to weak to intermolecular interactions of C-N stretching and N-H deformation in low molecular weight fragments. Amide A bands were recorded at 3337, 3395, 3377, and 3392 cm^−1^ in crude extracts of control, UAE, EAE, and EAE+UAE groups, respectively. Amide A bands for each fraction appeared in a wavenumber of 3420–3446 cm^−1^. Amide B bands appeared in the range of 2933–2938 cm^−1^ in all crude extracts, and from 2960 to 2967 cm^−1^ in fraction samples. Therefore, it can be determined that the structure was slightly changed in each fraction when compared with the crude extract using the FT-IR spectrum.

### 3.5. SDS-PAGE Analysis

The protein patterns of collagen hydrolysate in crude extracts and collagen fractions from the control, UAE, EAE, and EAE+UAE groups are presented in Figure 5. The crude extracts and ≥30 kDa fractions of the control and UAE groups showed wide bands of high intensities in the range of 37–250 kDa. As the molecular weight of the fractions decreased, bands above 50 kDa became less detectable. In particular, no bands were observed above 10 kDa for fractions of ≤3 kDa. In EAE and EAE+UAE, bands above 10 kDa were not observed in samples of all fractions. It can be inferred that the molecular weight of the EAE and EAE+UAE collagen peptides decreased due to enzymatic hydrolysis of intermolecular bonds. In a previous study, fragments >10 kDa were not detected in SDS-PAGE gels when Alcalase^®^ enzyme was used under the same conditions with this study [7]. On the other hand, collagen-related bands (α, β, γ) were faint in ≥30 kDa and crude extract fractions of the control and UAE groups, which is presumed to be due to the absence of acid extraction. The protein patterns of the control and UAE groups are similar to those obtained for controls in a similar experiment on peptide extraction from golden carp and grass carp skin [16,34]. It has been reported that 0.1 N NaOH can efficiently remove non-collagenous proteins without loss of collagen and is less harmful to the environment. This should be considered as an alternative to acid-based extraction approaches for the production of collagen in the future. 

## 4. Conclusions

Enzyme pretreatment combined with ultrasound significantly enhanced the extraction efficiency of collagen hydrolysate. Among the extracted collagens, EAE+UAE showed the highest total yield and yield of fragments with a molecular weight of <30 kDa. The collagen fractions from the control and UAE groups of ≤3 kDa exhibited the highest values for antioxidant activity, as measured using TEAC, FRAP, ORAC and inhibition of NO production by LPS-stimulated macrophages. The structural properties of extracted peptides were studied; peptides from the EAE and EAE+UAE groups were found to be the most effectively hydrolyzed. These findings support the use of a combination of enzymatic and ultrasound assisted extraction to produce bioactive collagen hydrolysate efficiently and in an environmentally friendly manner. Collagen hydrolysate fragments ≤3 kDa demonstrated physiochemical properties that support their use as an ingredient in functional food, pharmaceutical, and medical industries, and further work to improve their yield is needed.

## Figures and Tables

**Figure 1 antioxidants-11-02112-f001:**
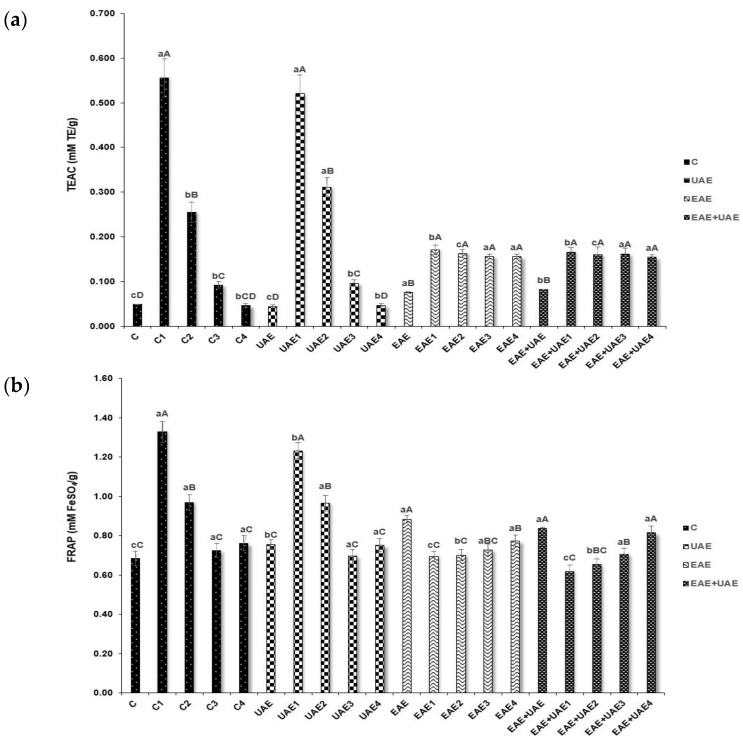
Antioxidant properties of collagen hydrolysate fractions extracted from APS. (**a**) TEAC, (**b**) FRAP, (**c**) ORAC. Results are calculated with means of three replications ± standard deviation. C, control; UAE, ultrasound assisted extraction method; EAE, enzyme assisted extraction method; EAE+UAE, combination of enzyme and ultrasound assisted extraction methods. 0; Crude extract, 1; ≤3 kDa, 2; 3–10 kDa, 3; 10–30 kDa, 4; ≥30 kDa. Same lowercase letters represented no significant difference between extraction procedures within protein fractions, and same capital letters represented no significant difference between protein fractions within extraction procedure.

**Figure 2 antioxidants-11-02112-f002:**
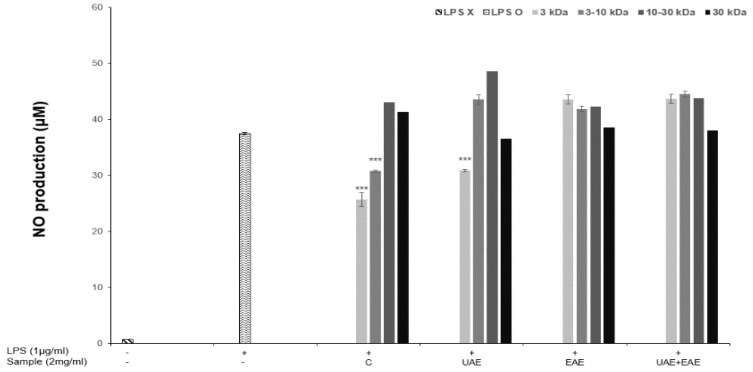
Inhibitory effect of collagen fractions extracted from APS on nitric oxide production in LPS-stimulated RAW 264.7 macrophage cells. C, control; UAE, ultrasound assisted extraction method; EAE, enzyme assisted extraction method; EAE+UAE, combination of enzyme and ultrasound assisted extraction methods. *** statistical significance of *p* < 0.001.

**Figure 3 antioxidants-11-02112-f003:**
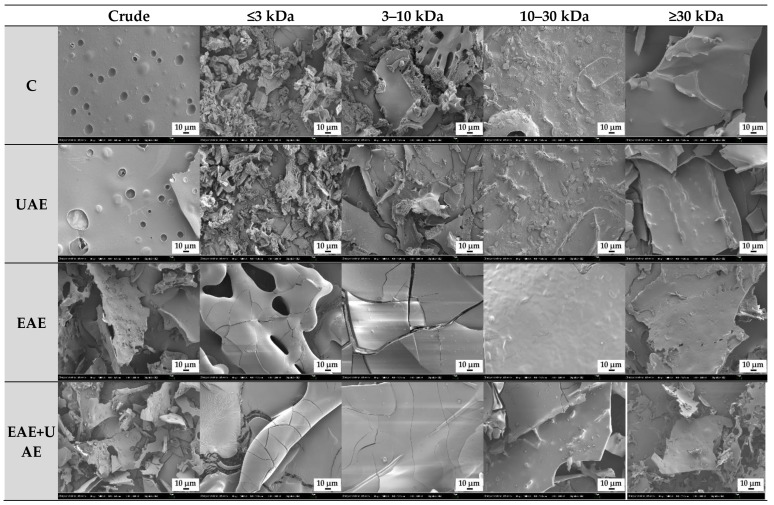
SEM image of collagen hydrolysate fractions extracted from Alaska pollack skin. C, control; UAE, ultrasound assisted extraction method; EAE, enzyme assisted extraction method; EAE+UAE, combination of enzyme and ultrasound assisted extraction methods (1000×).

**Figure 4 antioxidants-11-02112-f004:**
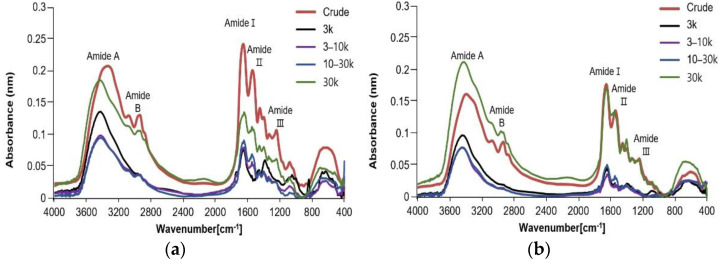
FTIR spectrum of collagen hydrolysate fractions extracted from APS. (**a**) Control; (**b**) Ultrasound assisted extraction method; (**c**) Enzyme assisted extraction method; (**d**) Combination of enzyme and ultrasound assisted extraction methods.

**Figure 5 antioxidants-11-02112-f005:**
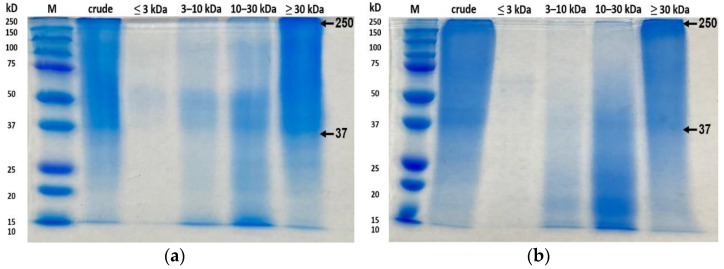
SDS-PAGE pattern of collagen hydrolysate fractions extracted from APS. (**a**) Control; (**b**) Ultrasound assisted extract; (**c**) Enzyme assisted extract (**d**) Combination of enzyme and ultrasound assisted extraction methods.

**Table 1 antioxidants-11-02112-t001:** The extraction yields of collagen hydrolysates fractions from Alaskan pollock skin (APS).

Sample ^(1),(2)^	Yield *** (%)
C0	31.00 ± 2.68 dA
C1	0.18 ± 0.14 cC
C2	0.57 ± 0.23 bC
C3	2.17 ± 0.93 bC
C4	21.16 ± 1.63 aB
UAE0	40.22 ± 1.29 cA
UAE1	0.39 ± 0.2 cC
UAE2	0.55 ± 0.27 bC
UAE3	3.16 ± 1.4 bC
UAE4	25.99 ± 0.69 aB
EAE0	62.66 ± 1.36 bA
EAE1	4.76 ± 0.65 bD
EAE2	11.62 ± 1.69 aC
EAE3	15.95 ± 1.64 aB
EAE4	14.74 ± 1.48 bBC
EAE+UAE0	65.33 ± 0.98 aA
EAE+UAE1	6.46 ± 1.17 aD
EAE+UAE2	12.90 ± 0.98 aC
EAE+UAE3	17.84 ± 0.34 aB
EAE+UAE4	11.73 ± 3.03 bC

The extraction yields are calculated with means of three replications ± standard deviation. ^(1)^ Control (C), ultrasound-assisted extract (UAE), Enzyme-assisted extract (EAE), Enzyme with ultrasound-assisted extract (EAE+UAE). ^(2)^ 0; Crude extract, 1; ≤3 kDa, 2; 3–10 kDa, 3; 10–30 kDa, 4; ≥30 kDa. Same lowercase letters represented no significant difference between extraction procedures within protein fractions, and same capital letters represented no significant difference between protein fractions within extraction procedure; *** statistical significance of *p* < 0.001.

## Data Availability

Data generated during this study are contained within the article.

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
