# Peer review of "Effects of Enzymatic- and Ultrasound-Assisted Extraction on Physicochemical and Antioxidant Properties of Collagen Hydrolysate Fractions from Alaska Pollack (Theragra chalcogramma) Skin"

_antioxidants, 2022, doi:10.3390/antiox11112112_

Round 1
Reviewer 1 Report
In this work, collagen hydrolysate fractions were extracted from Alaska pollock skin using enzymatic and ultrasound pre-treatment, both in isolation and in combination. The results are carefully demonstrated. However, minor revision is needed.
1. L60, in fact, there are many reports associated with the combinate methods of enzyme and ultrasound to extract collagen hydrolysate, such as: (1) Ultrasonic irradiation in the enzymatic extraction of collagen[J]. Ultrasonics sonochemistry, 2009, 16(5): 605-609. (2) VIDAL A R, Ferreira T E, Mello R O, et al. Effects of enzymatic hydrolysis (Flavourzyme®) assisted by ultrasound in the structural and functional properties of hydrolyzates from different bovine collagens[J]. Food Science and Technology, 2018, 38: 103-108. Therefore, the innovation of this work should be clearly addressed.
2. L77, how cold?
3. L95, provide the parameters conducted on the freeze dryer.
4. L202-295, Figure 3, the scale bars are not clear.
Author Response
"Please see the attachment"

Reviewer 2 Report
The manuscript provides an interesting study related to green processing and extraction of valuable wastes from commercial marine species. It is well presented and justified. I think it can be accepted for publication provided some aspects are performed that I think, could enhance its quality.
Abstract
It is not clear what extraction procedures were tested and compared (isolation and combination ?). Authors ought to express it in a better way.
Keywords
Include: microstructure.
Introduction
Line 60: Replace “are” by “is”.
Line 60: Perform the format of references.
Material and methods
Line 108: … allowed to react …
Lines 113-114: The word “samples” is ambiguous. Please clarify. Also in lines 125 and 134.
Line 143: Include the country.
Results
Table 1
All data are compared together. But two different factors (protein fraction and extraction procedure) are mixed. The authors ought to separate both factors. First, for the same protein fraction, a comparison among extraction procedures could be done (denoted by lowercase letters for example); secondly, for each extraction procedure, comparison among the different protein fractions (denoted by capital letters).
Figure 1
The same consideration as for Table 1. This way the discussion could be better focused on both factors separately.
Author Response
"Please see the attachment"
